# Heroes or Villains? The Dark Side of Charismatic Leadership and Unethical Pro-organizational Behavior

**DOI:** 10.3390/ijerph17155546

**Published:** 2020-07-31

**Authors:** Xue Zhang, Liang Liang, Guyang Tian, Yezhuang Tian

**Affiliations:** Harbin Institute of Technology, School of Management, Harbin 150001, China; 15b910027@hit.edu.cn (X.Z.); 18b910052@stu.hit.edu.cn (L.L.)

**Keywords:** charismatic leadership, unethical pro-organizational behavior, psychological safety, performance pressure

## Abstract

Although prior research has emphasized the disproportional contributions to organizations of charismatic leadership, an emerging line of research has started to examine the potentially negative consequences. In this paper, a theoretical framework was proposed for a study of unethical pro-organization behavior through psychological safety based on social information processing theory, which reveals the detrimental effect that charismatic leadership can have on workplace behavior. To explore this negative possibility, a time-lagged research design was applied for the hypotheses to be verified using 214 pieces of data collected from a service company in China. According to the results, unethical pro-organizational behavior was indirectly influenced by charismatic leadership through psychological safety. Moreover, when employees experienced high performance pressure, charismatic leadership was positively associated with unethical pro-organizational behavior through psychological safety. The implications of these findings were analyzed from the perspectives of charismatic leadership theory and organizational ethical activities to alter the unethical pro-organizational behavior.

## 1. Introduction

The detrimental effects of unethical behavior have been documented extensively and include detrimental effects on individual wellbeing, future careers, and organization survival. In general, it is believed among organizational researchers and managers that unethical behavior is excessively destructive or entirely driven by self-interest. Nevertheless, scholars have discovered that employees may perform unethical behavior for the interest of their company, such as lying to clients or hiding information from the public. To understand these unique phenomena, Umphress et al. proposed the construct of unethical pro-organizational behavior (UPB), which refers to activities conducted to potentially enhance the operation of the company, leaders, or members, yet breaches critical social values and damages the interests of external stakeholders [1]. Given the prevalence and negative consequences of this behavior, it is significant to understand when and why employees engage in UPB. Researchers have conducted studies on the antecedents of UPB based on social identity theory, social exchange theory, and social learning theory [2,3,4]. These researchers argued that UPB is significantly associated with leadership style [5,6,7], individual differences [8,9], personality, and values [10], as well as job characteristics [11]. Among these factors, leadership plays a crucial role in employees’ UPB. For instance, Graham et al. argued that leaders’ style and frames are significant factors in employees’ UPB [7]. Therefore, researchers have explored why various leadership styles play different roles in UPB. Notably, recent empirical findings implicating charismatic leadership suggest that unethical behavior may be invoked by leadership styles, which creates more risks and uncertainty in work environments [12]. Indeed, despite charismatic leadership being capable of producing strong positive effects on followers’ behavior in companies, it can also lead to adverse consequences [13]. Charismatic leadership is regarded as a dominant style of leadership in enterprises as a result of its signal values-based, symbolic, and emotion-laden style. Charismatic leadership is known to cope with cognitive and emotional challenges to produce positive outcomes in the workplace [14]. Fragouli (2018) proposed that the bright side of charismatic leadership can be eclipsed by its dark side, which causes detriment to the company [15]. Charismatic leaders are likely to build an egalitarian, non-exploitative, and altruistic organizational culture. Nevertheless, the behavior of charismatic leaders can increase the risk levels of the organization by introducing instability and uncertainty into decision-making processes [16]. Charismatic leaders inspire followers to take risks and motivate followers to achieve high corporate objectives [17,18,19]. As a result of being manipulated by charismatic leaders, staff members can be driven to perform an organizational mission or breach the ethical bottom line [20].

As such, this study concentrated on how charismatic leadership relates to UPB. Unfortunately, previous research related to the correlation between particular leadership and UPB has yielded conflicting outcomes and failed to clarify how charismatic leadership may lead to UPB in the workplace. As argued by Dang et al., a pressing issue in research is to determine how leadership facilitates UPB [21]. Moreover, this research will further contribute to theoretical knowledge by exploring the underlying psychological process of UPB, rather than exploring one single antecedent factor of UPB [1]. While social identification, social learning, and social exchange theory have been applied to account for the relationship between specific leadership and UPB [3,22,23], prior studies have failed to adequately explain why psychological safety is considered a form of interplay between leadership and unethical behavior. Therefore, drawing on social information processing, we provide a new perspective on UPB by examining the effect of charismatic leadership on motivating UPB through employees’ psychological safety. After controlling a set of control variables that would affect the variables of interest and their relationship, this study answers the call to examine a comprehensive model linking charismatic leadership to unethical behavior. Besides, we suggest that performance pressure can be a crucial factor in calibrating followers’ reactions toward leader influences. Specifically, the more performance pressure, the more a charismatic leadership will stimulate followers’ psychological safety, which will be positively associated with UPB; in contrast, the less performance pressure, the less a charismatic leadership will stimulate followers’ psychological safety, which will be positively associated with UPB. A model of hypothesized relationships is shown in Figure 1.

By applying social information processing to the context of UPB, we try to examine the mediated link between charismatic leadership and employee cognition and behaviors, aiming to make at least four notable contributions. First, we contribute to the UPB literature by identifying more antecedent factors of UPB. Through exploring the dark side of charismatic leadership, our research offers an alternative perspective on the antecedents of UPB. We suggest that employees are motivated to engage in UPB when they perceive that it is safe to take risks. Understanding charismatic leadership related to specific unethical behaviors provides precious insights into the cognitive processes underlying the motivation of ambivalent behaviors that have both unethical and good-intention elements.

Second, this research helps unpack the intricacies of UPB. Psychological safety explains the link between charismatic leadership and followers’ UPB, which complements the social information processing perspective. We invoke psychological safety as one reason why charismatic leadership may lead to followers’ UPB. Previous theory has limitations in predicting why and when employees may engage in unethical behaviors that benefit the company under the supervision of a charismatic leader; thus, we use theoretical perspectives on social information processes to advance understandings of the triggers of UPB.

Third, we develop hypotheses implicating performance pressure as an essential characteristic that can strengthen or attenuate the effects of charismatic leadership on UPB. Our results contribute to understanding the UPB phenomenon by exploring the roles of performance pressure in a field study. Collectively, this work provides significant evidence to suggest that followers engage in UPB as a function of the psychological safety conveyed by charismatic leaders, which is more likely to be felt among followers with high performance pressure.

Finally, our focus on the harmful effects of charismatic leadership on UPB provides a new perspective on charismatic leadership literature. To date, scholars have exclusively examined charismatic leadership effects on followers’ positive behavior, yet have not investigated how charismatic leadership may influence subordinate UPB. We thus answered the call from Bratton by highlighting that great leadership does not always lead to good outcomes [24]. Therefore, this study aimed to rely on social information process theory to advance theoretical knowledge by exploring the underlying mechanism linking charismatic leadership to UPB.

## 2. Theoretical Background and Hypotheses

### 2.1. Social Information Processing Theory

Social information processing theory (SIP) indicates that individuals intercept information from their immediate environment for the development of their attitudes, cognition, and behaviors [25]. At the workplace, leaders are one of the primary social environment sources from which members gather clues about attitudes and behaviors when uncertainty and ambiguity arise [26]. The principles of SIP are useful in explaining how charismatic leaders influence followers’ behavior through the interpretation of information. When confronted with charismatic leadership, followers may process the information provided by the charismatic leader and adjust their cognition and behaviors to the leadership environment accordingly. Charismatic leadership can convey to employees the social information that supervisors have confidence in the followers’ ability and excellent performance [27]. Influenced by this information, employees are inclined to interpret supervisors’ behaviors as supportive and a source of psychological safety. This channel is implied or assumed in discussions of charismatic leadership in many studies of leadership and its effects on followers. Thus, we submit that it is critical to test the degree to which the social information process affects followers’ UPB.

### 2.2. Charismatic Leadership and Psychological Safety

Psychological safety indicates the degree to which employees feel safe and confident in their abilities [28]. Psychological safety entails the perception that individuals are not scared of negative consequences to their self-impression, status, or career development [29]. Individuals are more likely to feel safe when they act within the boundaries of appropriate behavior [30]. That is, employees have a trusting and supportive relationship with their leaders and colleagues [29], and followers whose actions align with their leaders’ preferences are likely to feel more confident that they will be rewarded [31]. Concerning other corporate climate-related constructs, psychological safety is derived from interactions with leaders. Prior studies have highlighted that leadership is a significant influencing factor in psychological safety and revealed that leaders occupy a dominant position, that could shape followers’ psychological safety [32]. For organizational members, leaders are instrumental in determining what is part of their job and what is not [33]. For instance, Xu, Qin, Dust, and DiRenzo suggested that employees are less likely to feel psychological safety when they contradict the tendencies of their leaders, as they presume that their preferred approach to work will be perceived negatively [34]. Thus, we argue that charismatic leadership is crucial to enhancing followers’ psychological safety.

Charismatic leadership is referred to as using personal charm, attractiveness, and persuasive communication to exert influence on employees [35]. The social information process viewpoint is especially applicable to charismatic leaders, as their characters set high-performance expectations, instill hope and optimism, and gain trust and respect from their followers [27]. Charismatic leaders transform organizations and members in ways that are distinct from other leaders. They are capable of motivating followers to invest the most effort through articulating a vision for an organization’s future [16,36]. Charismatic leaders also inspire followers to pursue self-development [37] and lead to the satisfaction of followers. When working with charismatic leaders, followers usually feel more confident about themselves and their circumstances in the organization. Psychological safety is defined as a state in which individuals feel that it is safe to take risks when subject to many constraints to achieve corporate goals [38]. Based on SIP, if leaders can convey high confidence in the followers’ abilities, followers will feel free from potential threats or embarrassment resulting from mistakes. Thus, charismatic leadership is essential for the corporate climate by exhibiting charisma [39,40].

The literature on psychological safety has documented the essential role of leadership in fostering psychological safety by creating norms and guidelines. In prior literature, it was asserted that charismatic leaders are good at emphasizing the relation between effort and essential values, displaying confidence in the followers’ abilities, and communicating high-performance expectations by gaining trust and respect from followers [27]. Charismatic leadership develops a positive interface with the operating environment by assisting the company to improve performance. This is beneficial to the organizational climate, as employees can believe in each other without worrying about interpersonal risk, which is a significant feature of psychological safety [31,41,42,43]. When the leader exhibits some charismatic behavioral information, followers may interpret such information and infer that their risky behaviors are likely to be tolerated by the leader [44]. These arguments led to our prediction that charismatic leadership influences the psychological safety perceived by staff, and we hypothesized that:

**Hypothesis** **1**:*Charismatic leadership is positively related to followers’ psychological safety*.

### 2.3. Charismatic Leadership, Psychological Safety, and Unethical Pro-organizational Behavior

SIP indicates that charismatic leadership as a manager’s charming behavior was facilitated among followers through organizational skills. In scholarly research devoted to understanding the effect of charismatic leadership on follower cognition and workplace behaviors, prior research has primarily indicated positive relationships between charismatic leadership and corporate performance [18], knowledge sharing [45], and innovative behavior [19]. Nevertheless, these studies primarily considered the heroic aspects of charismatic leadership, with less attention devoted to its harmful effects [14,18], which makes it challenging to obtain a clear picture of the implications of charismatic leadership. To bridge this gap, and in response to Eisenbeib and Boerner’s call for research exploring the dark side of charismatic leadership [46], we believe that UPB in this context indicates a critical negative side-effect.

Despite the positive outcomes arising from charismatic leadership, researchers have recently proposed that employees may engage in UPB [5]. UPB creates a dilemma in which the interests of external stakeholders and customers are undermined while offering gains to the company. Prior studies demonstrated that UPB is partially motivated by various factors (e.g., organizational identity or commitment to the organization) and organizational identification mechanisms [6,47], yet have not examined the social information processing mechanism relating charismatic leadership to UPB.

To address this issue, the current study reveals that charismatic leadership is associated with UPB and that this relationship is mediated by psychological safety. First, charismatic leadership involves communicating a visionary mission and establishing high expectations for followers. This process is promoted through the leader, encouraging followers to exhibit considerable effort and emphasizing a collective identity. Followers who share a charismatic relationship with leaders are likely to identify with them and feel a heightened sense of collective identity and empowerment [48]. When leaders exhibit charisma, articulate an organizational vision, and describe the significance of achieving this vision, followers are more likely to feel safe when taking risks [13]. Thus, followers have a willingness to perform unethical behavior for the sake of the organization, to identify with the vision articulated by charismatic leaders [49,50].

Second, charismatic leaders are prepared to take high risks and engage in unconventional approaches to achieve the corporate vision. When followers trust that their leader has sufficient ability, they will feel comfortable in taking risks because of the belief that they will avoid punishment when risk-taking leads to unfavorable outcomes [51,52]. Indeed, psychological safety has been described as a critical affect-laden cognition that leads to unethical behavior [32]. Prior studies have confirmed the positive effect of psychological safety on unethical behaviors [53]. This suggests that charismatic leaders may be one of the predictors that cultivate followers’ confidence in their psychological safety, which in turn contributes to UPB. Thus, we predicted that psychological safety mediates the relationship between charismatic leadership and UPB among followers:

**Hypothesis** **2**:*Psychological safety will have a positive relationship with followers’ UPB*.

**Hypothesis** **3**:*Charismatic leadership will have a positive and indirect relationship with followers’ UPB through psychological safety*.

### 2.4. Moderating Effect of Performance Pressure

Individuals will perceive external information conveyed by leaders in different ways and respond differently to the same leadership behaviors. Previous studies suggested the significance of considering the role played by charismatic leadership through a contextual lens [13]. To gain an understanding of the contextual factors influencing followers’ behavior, we considered performance pressure as a moderator, which has been discussed in the literature on unethical behavior [54]. Performance pressure indicates the expectation that followers must deliver excellent performance outcomes [55,56]. Research on performance pressure has revealed negative consequences, in that it seems to increase poor ethical decision making [55,57,58] and stress [59].

We expect that performance pressure operates as a second-stage moderator, converging with psychological safety to facilitate complex behavior which violates moral norms to benefit organizations. Indeed, the combination of performance pressure and the risk-taking confidence encouraged by psychological safety contributes to a condition in which decisions to engage in unethical behavior for the interest of an organization have a higher likelihood of being encouraged by charismatic leadership. Performance pressure is accompanied by a belief that the current performance is insufficient for achieving what is required [60]. During an increase in performance pressure, the salience of consequences also increases [61]. Employees are aware that their efforts are related to consequences [62,63]. As suggested by Gutnick et al., performance pressure involves both high demands and high stakes [63]. Heightened performance pressure makes followers accountable for high-quality outcomes, which could lead to an individual achieving a riskier, potentially superior outcome [64]. When confronting a decision, followers who experience high performance pressure can take advantage of the safety of their circumstances to achieve the most beneficial outcome, even if it is considered unethical. As such, as an ethically laden concept, UPB should be more relevant to individuals experiencing higher performance pressure.

When facing high performance pressure, individuals who experience high psychological safety are more likely to conduct UPB, given that charismatic leaders emphasize collective identity, which may motivate followers to do whatever they can to accomplish organizational goals [65]. In particular, charismatic leaders are good at motivating followers to demonstrate considerable effort and behaviors to achieve enhanced performance [66]. When followers perceive performance gaps, they will seek to overcome these to get the job done [67]. The completion of tasks and attitudes toward the completion of work contribute to the generation of risk-taking. Psychological safety is described as a critical cognition in determining unethical behavior because of the interpersonal risks inherent in UPB, given that psychological safety provides a mechanism through which followers perceive the lower potential costs of engaging in UPB.

In contrast, when under low performance pressure, employees with high psychological safety have a lower likelihood of engaging in UPB. While psychological safety induced by charismatic leadership has the potential to prompt followers into engaging with UPB, followers with lower performance pressure are less prone to this effect. People with lower performance pressure consider the moral import of their decisions and are motivated to behave ethically, thereby eliminating the ethicality associated with unethical acts. Thus, we proposed a second-stage moderation of performance pressure on the psychological safety–UPB relationship:

**Hypothesis** **4**:*Performance pressure will moderate the relationship between psychological safety and UPB, such that the relationship will be stronger when performance pressure is higher, rather than lower*.

Taken together, it is logical to predict that psychological safety will conditionally influence the strength of the indirect relationship between charismatic leadership and UPB. It is expected that high performance pressure strengthens the indirect relationship by enhancing the mediating effect of psychological safety between charismatic leadership and UPB. Hence, by facilitating psychological safety, employees with high performance pressure tend to engage in UPB under the supervision of charismatic leaders. Thus, we propose a moderated mediation model to explain the effect of charismatic leadership on employees’ UPB. We hypothesize that:

**Hypothesis** **5**:*Performance pressure will moderate the indirect relationship between charismatic leadership and UPB through psychological safety, such that the relationship will be stronger when performance pressure is higher, rather than lower*.

## 3. Methodology

### 3.1. Participants and Procedure

This study was performed in a large service company located in Ji Lin city, northeast China. This company has about 1000 employees and provides skiing services to tourists. The CEO and human resource department distributed research announcements to employees. The participants in this study comprised 300 full-time working adults who were selected at random from a list of employees in the company. The service industry presents some unique interactions between employees and customers that make it particularly suitable for our research purpose. Previous research has demonstrated that such a workplace environment is ideal for examining third-party oriented behaviors and UPB [9,68]. Thus, we selected this sample to test our hypotheses by focusing on a single industry in the China context. In order to avoid common method bias, this study adopted a multi-period data collection method where participants were asked to answer the questionnaires throughout the whole of February in 2019. Previous research suggests that time separation between predictor and criterion variables can reduce standard method variance bias by decreasing consistency motifs and demand characteristics [69]. Thus, we used three separate surveys with intervals of two weeks between each survey. A time lag of two weeks is consistent with published work examining the influence of leadership on UPB [70].

We collected data from eight departments and relied on convenience and snowball sampling. In the first survey (T1), with the help of the CEO and human resource manager (HR), we asked followers to fill in the questionnaire about gender, age, education, work tenure, and charismatic leadership. Two weeks later, at Time 2, participants rated psychological safety. Two weeks after this, at Time 3, respondents provided ratings on UPB. There were 276 respondents in the Time 1 survey (92% response rate), 232 participants in the Time 2 survey (84.05% reaction rate), and 226 respondents in the Time 3 survey (97.41% reaction rate). Among them, nine did not provide a complete response in the whole study. Furthermore, the respondents who incorrectly responded to the items and those from the three waves that could not be matched were removed from the analysis, with 214 effective participants remaining in the final sample. Among these final respondents, 62.62% were men, and 46.26% held a bachelor’s degree. Most participants were aged 25 to 30 years (43.46%), with 37.85% having more than three years of work experience. The participants’ departments were, respectively, customer reception (12.56%), ski marketing (25.61%), amusement park marketing (17.12%), security (1.51%), after-sale services (4.33%), membership card marketing (27.92%), transport services (5.08%), and ski training (5.87%). The sample demographic frequencies are shown in Table 1.

### 3.2. Measurements

Charismatic leadership: Charismatic leadership was assessed by applying a 25-item scale proposed by Conger and Kanungo [71] on a five-point Likert scale (1 = “Strongly disagree” to 5 = “Strongly agree”). The 25 items were divided equally into five subscales that measure the strategic vision and articulation, personal risk, sensitivity to the environment, sensitivity to member needs, and unconventional behavior. The sample items were “appears to be a skillful performer when presenting to a group” and “recognizes the abilities and skills of other members in the organization.” Because this study focusses on the overall effect of charismatic leadership rather than the different effects of subdimensions, we summed the 25 items to arrive at an overall index of charismatic leadership following previous studies. The Cronbach’s alpha was 0.90, and the KMO & Bartlett’s test was 0.87 (*p* < 0.01).

Psychological safety: Psychological safety was measured using three items from Liang and Farh’s scale [72]. Items were rated on a five-point Likert scale (1 = “Strongly disagree” to 5 = “Strongly agree”). We selected these three items because they are typical cognitions in the service industry and have more representatives with a higher loading score (more than 0.60) than the other two items on the original scale. We also invited Ph.D. students to select items that suited the organizational context and assess the final scale to guarantee content validity. The pre-test with the 73 samples showed good reliability and construct validity (the loadings of the items ranged from 0.62 to 0.84, and the Cronbach’s α was 0.71). The sample items were “I can express my true feelings regarding my job in my organization” and “I can freely express my thoughts in my organization.” The Cronbach’s α was 0.79, and the KMO & Bartlett’s test was 0.70 (*p* < 0.01).

Unethical pro-organizational behavior: The respondents were required to assess UPB by applying Umphress et al.’s six-item scale [1]. Items were rated on a five-point Likert scale (1 = “Strongly disagree” to 5 = “Strongly agree”). A sample item was “I exaggerated the truth about my company’s products or services to customers and clients.” The Cronbach’s alpha was 0.74, and the KMO & Bartlett’s test was 0.74 (*p* < 0.01).

Performance pressure: Performance pressure was examined with a three-item scale proposed by Rubin, Dierdorff, and Brown [73]. Items were rated on a five-point Likert scale (1 = “Strongly disagree” to 5 = “Strongly agree”). Sample items were “There is a great deal of pressure to perform here.” The Cronbach’s α was 0.74, and the KMO & Bartlett’s test was 0.70 (*p* < 0.01).

Control variables: Based on the recommendations of previous studies [21], demographic characteristics may exert influence on participants’ propensity to engage in UPB. We used age, gender, education, and work tenure to control for their potentially spurious effects, as the findings of Kish-Gephart et al. revealed a week correlation between gender and age and unethical actions [74]. Thus, information on followers’ gender, age, education, and work tenure was controlled.

### 3.3. Data Analysis

In this study, we measured all constructs at the individual level and tested the hypothesized measurement model using Mplus 7.4. Next, the regression-based approach was obtained for testing each hypothesis in the theoretical model. Following Preacher et al., we examined the indirect effects and moderation hypotheses using a bootstrapping procedure with 5000 replications to derive CIs in SPSS 22.0 PROCESS [75].

## 4. Results

### 4.1. Tests of Measurement Models

A range of confirmatory factor analyses (CFA) was conducted using Mplus 7.4 to verify how distinctive the variables would be [76]. Four indices were used to evaluate the goodness of fit: the chi-square statistic (χ^2^), comparative fit index (CFI), Tucker–Lewis index (TLI), and the root mean square error of approximation with associated 90% confidence intervals (RMSEA). CFI ≥ 0.90 and RMSEA ≤ 0.06 indicate a model’s acceptable fit to the data. The overall CFA results confirmed that the proposed four-factor model fitted the data excellently (χ^2^ = 145.72, df = 49, χ^2^/df = 2.97, CFI = 0.91, TLI = 0.90, RMSEA = 0.08), as shown in Table 2. To examine whether common bias had affected our results, Harman’s one-factor model test was performed, inclusive of the 37 items obtained from the same source (i.e., followers) in one model, and a comparison was performed between its model fit indices and the measurement model. The results showed that the one-factor model with a combination of all items was a poor fit for the dataset (χ^2^ = 917.08, df = 64, χ^2^/df = 14.33, CFI = 0.42, TLI = 0.25, RMSEA = 0.22). Therefore, we believe that common method variance did not have a significant effect on our data.

### 4.2. Hypotheses Testing

The analyses were modeled in SPSS 20.0 (SPSS Inc, Chicago, IL, USA) using hierarchical regression. Table 3 presents the means, standard deviations, and correlations among the researched variables. Consistent with the predictions, charismatic leadership is positively related to psychological safety (γ = 0.16, *p* < 0.01). In addition, psychological safety is positively associated with UPB (γ = 0.20, *p* < 0.01).

As displayed in Table 4, Model 2, the results indicated that charismatic leadership had a positive relationship with psychological safety (γ = 0.21, *p* < 0.05). Thus, Hypothesis 1 was supported. As displayed in Model 4, the results indicated that psychological safety had a positive relationship with UPB (γ = 0.23, *p* < 0.01). Thus, Hypothesis 2 was supported. For Hypothesis 3, the 95% bias-corrected bootstrap confidence intervals were determined for the assumed indirect effect from 5000 bootstrap samples. As shown in Table 5, the correlation between charismatic leadership and UPB via psychological safety was significant (γ = 0.05, 95% CI [0.01, 0.12]), which is a small but significant effect. Thus, Hypothesis 3 was supported.

Hypothesis 4 proposed that performance pressure would moderate the relationship between psychological safety and UPB, such that this positive relationship would be stronger in the presence of high (vs. low) performance pressure. Specifically, performance pressure had a significant moderation effect on the relationship between psychological safety and UPB (γ = 0.14, *p* < 0.05) in Table 4, Model 6. We plotted this moderation effect in Figure 2. Tests of simple slopes (at +/−1 SD of performance pressure) indicated that when performance pressure was high, psychological safety was significantly positively related to UPB (b = 0.33, t = 4.67, *p* < 0.01). While the simple slope computed at one standard deviation below the mean, the relationship between psychological safety and UPB was nonsignificant (b = 0.05, t = 0.53, ns). These results suggest that the moderation effect was statistically significant when performance pressure was high. Therefore, Hypotheses 4 was supported.

We analyzed the contingent indirect effect according to suggestions by Preacher, Rucker, and Hayes [75]. According to Table 6, in the case of higher performance pressure, the indirect effect of charismatic leadership on UPB through psychological safety was of significance (estimate = 0.07, CI [0.01, 0.16]). When performance pressure was low, the indirect effect of charismatic leadership on UPB through psychological safety was insignificant (estimate = −0.01, CI [−0.06, 0.05]). The overall indirect effect of charismatic leadership on UPB through psychological safety was also significant (estimate = 0.06, CI [0.01, 0.15]). Moreover, the indirect effect was stronger at higher levels of performance pressure than at lower levels (differences = 0.08, CI [0.01, 0.17]). Thus, Hypothesis 5 was supported. These effects are illustrated in Figure 3.

Note: PP = performance pressure. The indirect effect displays positivity and significance when performance pressure is equal to or greater than 3.50 and equal to or less than 4.14 (on a five-point scale).

## 5. Discussion

In this study, SIP was applied to investigate when and why employees perform unethical behavior for the interests of their company. Departing from prior research examining the heroic aspects of charismatic leadership, we explored why and when charismatic leadership leads to adverse outcomes, such as UPB. Our predictions revealed that charismatic leadership might provide psychological safety to subordinates and lead to UPB. We also explored the possibility that some followers with intense performance pressure may have a higher likelihood of engaging with UPB.

### 5.1. Theoretical Implications

This research has several necessary implications for leadership and ethical literature. First, we have underscored the role of charismatic leadership in boosting unethical behavior for the benefits of the organization. Psychological safety was proposed as one of the fundamental mechanisms connecting charismatic leadership and UPB. Our findings confirmed the theory that charismatic leadership enhances psychological safety, which results in the stamina required for UPB. It is significant to understand that charismatic leadership does not necessarily lead to positive outcomes [77]. Charisma is not a God-given characteristic and may be used for wrong and tragic ends. In particular, charismatic leadership can enhance the psychological safety of followers and inspire them to achieve organizational goals by taking risks, such as conducting unethical behavior for the benefit of the organization. As such, charismatic leadership is not necessarily good and does not necessarily lead to positive outcomes.

Second, our study has revealed the critical preconditions of the relationship between charismatic leadership and UPB through psychological safety, with our outcomes demonstrating that a substantial performance pressure amplifies this positive relationship. A psychologically safe environment is supposed to enhance employees’ wellbeing. Previous research has suggested that psychological safety facilitates positive appraisals of work and life and plays a decisive role in wellbeing [78]. This finding suggests that leaders’ charisma mainly exerts influence on unethical behavior for the benefits of the organization when followers experience high performance pressure and do not fear taking risks, which is accounted for by the fact that, in this circumstance, followers are motivated by their leader’s charismatic capabilities to endeavor to achieve organizational goals.

Third, we considered why a relationship might exist between charismatic leadership and UPB through SIP. Prior research has typically been reliant on fundamental tenets from social learning theory [79] or social exchange theory [80] to examine the antecedents of UPB. While admitting that these theories may explain the increase of UPB, we held the view that the effects of charismatic leadership on UPB will be more significant and enduring when psychological safety is formed through interpreting information from leaders and the environment.

### 5.2. Practical Implications

Our findings have several implications for organizations and practitioners. UPB is undertaken with good intentions, yet it is detrimental in the workplace. Our findings provide evidence that performance pressure can reinforce the relationship between charismatic leadership and UPB through psychological safety. Therefore, decision-makers requiring employees to achieve high performance should be aware of this dilemma and the management of employee UPB. Perhaps it is impractical to alleviate performance as organizations seek to maintain competitive advantage; however, managers can carefully consider how to uphold performance requirements while emphasizing the importance of ethical values. For instance, performance anticipation should be fulfilled with moral standards as the bottom line for how performance is achieved. Moreover, the infrastructure that staff members navigate in their performance must be linked to the company’s moral practices [57,70]. Leaders act as a crucial driver of employee ethical behavior and can adapt activities to promote ethical values and information to followers and reduce employees’ UPB.

### 5.3. Limitations and Future Directions

Despite the research contributions, this research also involved some limitations. First, we assessed all our hypotheses with self-reporting measures and raised concerns about common method bias. Future research can be conducted on how to reduce common method bias by adopting experimental, longitudinal, or quasi-experimental designs and including data from other sources, such as the assessments of UPB by leaders or coworkers. Second, charismatic leadership has two dimensions; we measured global charismatic leadership in this study. Future studies could focus on the effects of the two types of charismatic leadership on UPB to improve the research design and offer a more comprehensive understanding of charismatic leadership functions.

Despite these limitations, our research provides the basis for further study. The results suggest charismatic leadership as a means through which organizations can increase psychological safety in taking risks, thus reinforcing employees’ UPB. Researchers could consider other clues that may influence staff members’ motivation to participate in UPB. For instance, researchers have found that particular emotions may motivate unethical behavior. It would be useful for researchers to consider the effect of charismatic leadership from various perspectives. While our research focused on charismatic leadership from the perspective of individuals, it may be informative to evaluate charismatic leadership at the team level. A further study exploring the diverse nature of charismatic leadership may help account for how companies mitigate the harmful effects of charismatic leadership for enterprises.

We collected the data from a single industry and a single cultural context. The study was collected in China and may limit the generalizability of our findings. Previous studies have demonstrated that, traditionally, the Chinese have a strong spirit of sacrifice for the sake of collective interests [81]. The Chinese may treat UPB as a more appropriate way to repay their supervisors in response to charismatic leadership. Thus, the relationship between charismatic leaders and UPB may be stronger in Chinese firms than in Western companies. Therefore, we suggest that future studies verify the relationship between charismatic leadership and UPB in non-Chinese culture. Although our findings might probably be repeated in other industries, given the similarity of sale and service jobs in different industries, we cannot completely ensure the generalizability of our results to other industries. Future research may investigate more industries to ascertain the generalizability of our findings.

Finally, though our findings shed light on the role of charismatic leadership, the broader role of leadership and organizational culture remains unclear. Further research is needed to focus on the effects of other types of leadership in the organization on UPB. This might contribute to a more comprehensive knowledge about the functioning of leadership. Similarly, it may be worthwhile to consider additional mediating mechanisms that may transfer the effects of leadership on UPB. Our results showed that social information processing translated charismatic leadership’s effects into UPB. We did not consider the joint effects of leadership and moral judgments on UPB, and future research should consider how these relationships and moral cognition jointly contribute to the motivation to engage in UPB. Although we relied on theory to present directional hypotheses, field study is not ideal for establishing causal direction. Future research should use a combination of experimental research and repeated-measures longitudinal designs to establish the robustness of the effects.

## 6. Conclusions

Based on SIP, this study has proposed and validated a theoretical model on the relationship between charismatic leadership, psychological safety, performance pressure, and UPB. Following an analysis of 214 pieces of data gathered from a company in China, the results revealed a positive, indirect effect between charismatic leadership and UPB via psychological safety, and a moderating effect of performance pressure. Further, we concluded that charismatic leadership might alleviate employees’ UPB. Future research can explore further ways to reduce UPB from the perspective of social information processing.

## Figures and Tables

**Figure 1 ijerph-17-05546-f001:**
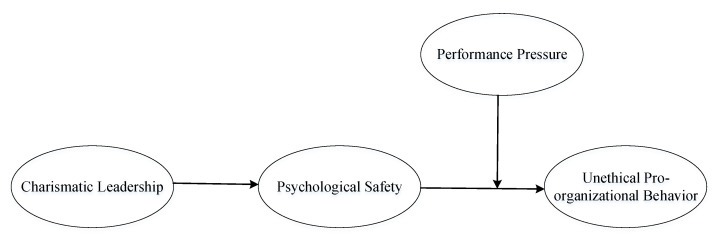
Theoretical Model.

**Figure 2 ijerph-17-05546-f002:**
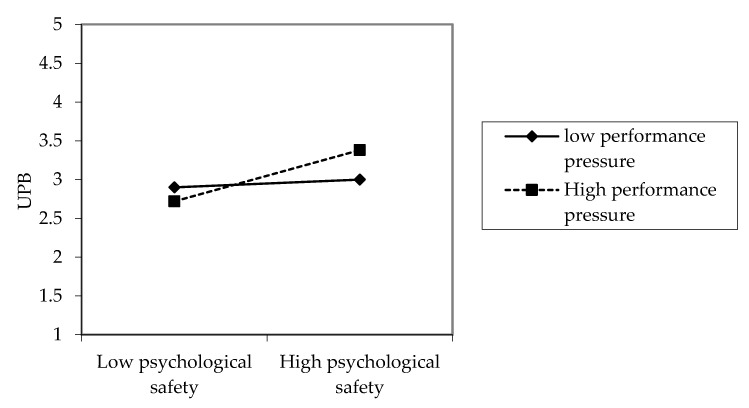
Moderating Effect of Performance Pressure.

**Figure 3 ijerph-17-05546-f003:**
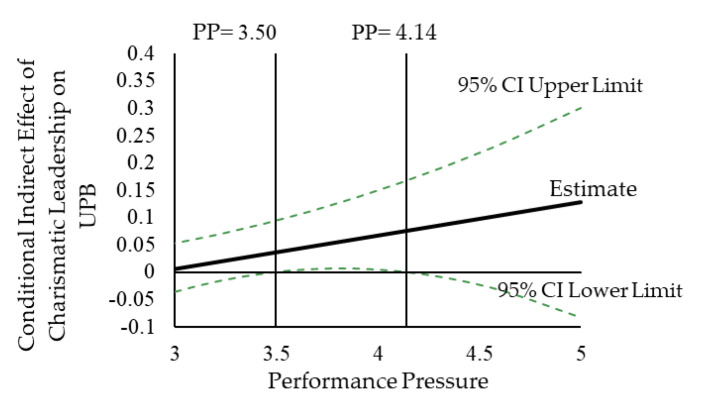
Conditional Indirect Effect of Charismatic Leadership and UPB via Performance Pressure.

**Table 1 ijerph-17-05546-t001:** Sample Demographic Frequencies.

Demographic		Frequency	Percent
Gender	Male	134	62.62
	Female	80	37.38
Age	<25 years	92	42.99
	26–35 years	93	43.46
	36–45 years	18	8.41
	46–55 years	11	5.14
Education	high school	23	10.75
	junior college	86	40.19
	bachelor	99	46.26
	master	3	1.40
Tenure	<1 year	70	32.71
	1–2 years	28	13.08
	2–3 years	35	16.36
	3–4 years	26	12.15
	>5 years	55	25.70

**Table 2 ijerph-17-05546-t002:** Results of confirmatory factor analysis.

Model	χ^2^	df	χ^2^ /df	CFI	TLI	RMSEA
4-factor model ^a^	145.72	49	2.97	0.91	0.90	0.08
3-factor model ^b^	235.93	55	4.29	0.78	0.77	0.11
2-factor model ^c^	465.24	58	8.02	0.54	0.51	0.16
1-factor model ^d^	917.08	64	14.33	0.42	0.25	0.22

Note: ^a^ charismatic leadership, psychological safety, performance pressure, UPB; ^b^ charismatic leadership + psychological safety, performance pressure, UPB; ^c^ charismatic leadership + psychological safety + performance pressure, UPB; ^d^ charismatic leadership + psychological safety + performance pressure + UPB.

**Table 3 ijerph-17-05546-t003:** Means, Standard Deviations, and Correlations among Variables.

Variable	M	SD	1	2	3	4	5	6	7
1. Gender ^a^	1.37	0.48	–						
2. Age ^b^	1.76	0.81	−0.08	–					
3. Education ^c^	2.53	2.22	0.16 *	−0.04	–				
4. Work tenure ^d^	2.84	1.61	−0.03	0.64 **	0.09	–			
5. Charismatic leadership	2.54	0.68	0.07	−0.01	0.07	0.09	–		
6. Psychological safety	4.03	0.45	−0.05	−0.06	0.04	0.01	0.16 *	–	
7. Performance pressure	3.34	0.56	−0.16 *	−0.07	0.03	−0.03	0.12	0.02	–
8. UPB	3.50	0.57	−0.07	0.08	−0.09	−0.02	0.09	0.20 **	0.05

Note: *N* = 214. * *p* < 0.05, ** *p* < 0.01 (two-tailed); ^a^ 1 = male; 2 = female; ^b^ 1 = less than 25 year old; 2 = 26–35 year old; 3 = 36–45 year old; 4 = 46–55 year old; ^c^ 1 = high school; 2 = junior college; 3 = bachelor; 4 = master; ^d^ 1 = less than 1 year; 2 = 1–2 year; 3 = 2–3 year; 4 = 3–4 year; 5 = more than 5 year.

**Table 4 ijerph-17-05546-t004:** Regression Results.

Independent Variable	Psychological Safety		UPB			
	M1	M2	M3	M4	M5	M6
Gender	−0.06	−0.07	−0.07	−0.05	−0.04	−0.07
Age	−0.08	−0.07	0.12	0.13	0.14	0.13
Education	0.01	0.01	−0.02	−0.02	−0.03	−0.02
Work tenure	0.02	0.01	−0.05	−0.05	−0.05	−0.05
Charismatic leadership		0.21 *	0.18	0.13	0.12	0.10
Psychological safety				0.23 **	0.23 **	0.19 *
Performance pressure					0.05	0.05
Psychological safety * performance pressure						0.14 *
R^2^	0.01	0.04	0.03	0.04	0.07	0.10
∆R^2^	0.01	0.03 **	0.03	0.03 *	0.01	0.03 **

Note: *N* = 214. * *p* < 0.05, ** *p* < 0.01 (two-tailed). Standardized coefficients are presented for the linear regression.

**Table 5 ijerph-17-05546-t005:** Indirect Effects of Charismatic Leadership on UPB.

Hypothesis Indirect Effect	Mediator	Indirect Effect	SE	95% Confidence Interval
Hypothesis 2	Psychological safety	0.05 **	0.03	[0.01, 0.12]

Note: *N* = 214. ** *p* < 0.01 (two-tailed).

**Table 6 ijerph-17-05546-t006:** Conditional Indirect Effects.

Variables	UPB		
	Estimate	SE	95% CI
Low performance pressure	−0.01	0.03	[−0.06, 0.05]
High performance pressure	0.07 **	0.04	[0.01, 0.16]
Index of moderated mediation	0.06 **	0.04	[0.01, 0.15]
Indirect effect differences	0.08 **	0.04	[0.01, 0.17]

Note: *N* = 214. ***p* < 0.01 (two-tailed).

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
