# Peer review of "Heroes or Villains? The Dark Side of Charismatic Leadership and Unethical Pro-organizational Behavior"

_ijerph, 2020, doi:10.3390/ijerph17155546_

Round 1

Reviewer 1 Report

The research model is interesting, investigating mediators and moderators that make charismatic leadership a potential issue for organizations. Implications are relevant at both theoretical and practical level.

In which type of company was the data collected (which industry)? Do you think that results can be generalized across industries? Maybe openly stating the sector in which the company operates, and discussing this in the limitations section can provide a more accurate framing of the results.

On the same line, data was collected only in China: there is consolidated evidence of cross-cultural differences in ethical behavior and thought patters. Cross-cultural ethics should at least be mentioned in the implications.  

How is this research relevant to a journal of environmental research and public health? Wouldn’t it be worth adding a paragraph explaining the contextualization of the implications of your study for public health (i.e., workplace mental health)?

Why did you collect the data over time, splitting the questionnaire in 3 parts administered two weeks apart?

Line 274 you state that “Psychological safety was measured using three items modified”: could you provide more details on how and why the items were modified?

Line 166: “Hypothesis 1” should be bold

Table 2 is cut across 2 pages making it very difficult to read.

There is a published paper with a similar title, but the authors do not cite it, which seems a bit strange:

DeCelles, K. A., & Pfarrer, M. D. (2004). Heroes or villains? Corruption and the charismatic leader. Journal of Leadership & Organizational Studies11(1), 67-77

It was a pleasure having the chance to read your manuscript. Good luck!

Author Response

Point 1: The research model is interesting, investigating mediators and moderators that make charismatic leadership a potential issue for organizations. Implications are relevant at both theoretical and practical levels.

Response:

Thank you very much for your constructive feedback and guidance. We have made further revisions to the paper in response to your thoughtful comments. Below we describe these revisions point-by-point.

Point 2: In which type of company was the data collected (which industry)? Do you think that results can be generalized across industries? Maybe openly stating the sector in which the company operates, and discussing this in the limitations section can provide a more accurate framing of the results.

Response: Thanks for asking this question and prompting us to explain our argument better. In order to make the data collection process clearer, we revise the Participants and Procedure and Limitation section in the updated manuscript. The revised paragraph is as follows (p.6-7,11):

“This study was performed in a large service company located in Ji Lin city, northeast China. This company with about 1000 employees providing skiing services to tourists. The CEO and human resource department distributed research announcements to employees. The participants in this study comprised 300 full-time working adults who were selected at random from a list of employees in the company. The service industry presents some unique interactions between employees and customers that make it particularly suitable for our research purpose. Previous research has demonstrated that such a workplace environment is for examining third-party oriented behaviors and UPB [9,68]. Thus, we selected this sample to test our hypotheses by focusing on a single industry in China context. In order to avoid common method bias, this study adopted a multi-period data collection method where participants were asked to answer the questionnaires throughout the whole of February in 2019. Previous research suggests that time separation between predictor and criterion variables can reduce common method variance bias by decreasing consistency motifs and demand characteristics [69]. Thus, we used three separate surveys with intervals of two weeks between each survey. A time lag of two weeks is consistent with published work examining the influence of leadership on UPB [70].”

 “We collected the data from a single industry and a single culture context. The study was collected in China and may limit the generalizability of our findings. Previous studies have demonstrated that traditional Chinese have strong spirits of sacrifice for the sake of collective interests [81]. Chinese may treat UPB as a more appropriate way to repay their supervisors in response to charismatic leadership. Thus, the relationship between charismatic leaders and UPB may be stronger in Chinese firms than in western companies. Therefore, we suggest that future studies verify the relationship between charismatic leadership and UPB in non-Chinese culture. Although it is probable that our findings might be repeated in other industries in view of the similarity of sale and service jobs in different industries, we cannot completely ensure the generalizability of our results to other industries. Future research may investigate more industries to ascertain the generalizability of our findings.”

Point 3: On the same line, data were collected only in China: there is consolidated evidence of cross-cultural differences in ethical behavior and thought patterns. Cross-cultural ethics should at least be mentioned in the implications.  

Response: Thanks for prompting us to explain our theoretical arguments better. We revise the argument of the Limitation section to make it clear. The revised paragraph is as follows (p.11):

“We collected the data from a single industry and a single culture context. The study was collected in China and may limit the generalizability of our findings. Previous studies have demonstrated that traditional Chinese have strong spirits of sacrifice for the sake of the collective interests [81]. Chinese may treat UPB as a more appropriate way to repay their supervisors in response to charismatic leadership. Thus, the relationship between charismatic leaders and UPB may be stronger in Chinese firms than in western companies. Therefore, we suggest that future studies to verify the relationship between charismatic leadership and UPB in non-Chinese culture. Although it is probable that our findings might be repeated in other industries in view of the similarity of sale and service jobs in different industries, we cannot completely ensure the generalizability of our results to other industries. Future research may investigate more industries to ascertain the generalizability of our findings.”

Point 4: How is this research relevant to a journal of environmental research and public health? Wouldn’t it be worth adding a paragraph explaining the contextualization of the implications of your study for public health (i.e., workplace mental health)?

Response:

Thanks for encouraging us to strengthen our theoretical arguments. We revise the argument of the Theoretical contribution section to make it clear. The revised paragraph is as follows (p.10):

“A psychologically safe environment is supposed to enhance employees’ wellbeing. Previous research has suggested that psychological safety facilitates positive appraisals of work and life, and plays a positive role in wellbeing [78].”

Point 5: Why did you collect the data over time, splitting the questionnaire in 3 parts administered two weeks apart?

Response: Thanks for providing us an opportunity to clarify your concerns. The revised paragraph is as follows (p.7):

“In order to avoid common method bias, this study adopted a multi-period data collection method where participants were asked to answer the questionnaires throughout the whole of February in 2019. Previous research suggests that time separation between predictor and criterion variables can reduce common method variance bias by decreasing consistency motifs and demand characteristics [69]. Thus, we used three separate surveys with intervals of two weeks between each survey. A time lag of two weeks is consistent with published work examining the influence of leadership on UPB [70].”

Point 6: Line 274 you state that “Psychological safety was measured using three items modified”: could you provide more details on how and why the items were modified?

Response: Thanks for providing us an opportunity to clarify your concerns. We have revised the measurement as follows (p.7).

“Psychological safety: Psychological safety was measured using three items from Liang and Farh’s scale [72]. We selected these three items because they are typical cognitions in the service industry. The sample items are ‘I can express my true feelings regarding my job in my organization’ and ‘I can freely express my thoughts in my organization’.”

Point 7: Line 166: “Hypothesis 1” should be bold

Response: Thanks for your suggestion. We have revised the hypotheses as you suggested.

Point 8: Table 2 is cut across 2 pages making it very difficult to read.

Response: Thanks for your suggestion. We have revised the tables to make it clear.

Point 9: There is a published paper with a similar title, but the authors do not cite it, which seems a bit strange:

DeCelles, K. A., & Pfarrer, M. D. (2004). Heroes or villains? Corruption and the charismatic leader. Journal of Leadership & Organizational Studies11(1), 67-77

Response: Thanks for your suggestion. We have added the reference as [14].

Point 10: It was a pleasure having the chance to read your manuscript. Good luck!

Response: We highly appreciate the time and effort you have committed to reviewing our paper. Our paper has improved substantially because of your comments. Thank you very much!

Reviewer 2 Report

This manuscript reports on an interesting study that explores charismatic leadership and unethical pro-organizational behavior. I applaud the aspirations represented in this paper. However, both formal and content aspects of the manuscript must be revised.

I hope the suggestions I give below will support you in advancing your research efforts on this topic. Following are my specific comments on this paper.

Introduction Review Points

  1. There is a clear statement of the purpose in the introduction. However, the information should be improve.
  2. The aim of the study requires a more detailed analysis of the literature on the subject to justify the theoretical model they propose. In addition, it also needs to develop more hypotheses.

Method Review Points

  1. The sample works in only one company. Is it a case study? I think that the study has a questionable sampling method and need to be improved.
  2. Appropriate identification of participants is critical and this study. This paper does not adequately described participants. Moreover, the authors should described each of the samples or, if they do intrasubject analyses, they should include only the participants who are at all times.
  3. The authors should detail the type of sampling, why this type of sampling is the best, what risks it has for its validity and how they have been solved so that they do not affect the result obtained.
  4. The scales are not correctly described
  5. The control variables are not justified in the introduction. Moreover, there are no hypotheses about them.
  6. There are not subsection: ‘data analysis’

Results Review Points

  1. It doesn't explain what the test of measurement models is for. Moreover, it does not indicate with which scales and in which time it has been carried out. It presents some results that we do not know what they correspond to
  2. The authors should review all the results. There are many inaccuracies. For example, include the mean and standard deviation of gender and education. Both are categorical variables.
  3. In addition, the correlations are very low and practically non-existent. This means that perhaps the hypotheses are not worth testing. For example, the correlation between charismatica leadership and psychological safety is 0.16. This is a very low positive correlation. Therefore, even if hypothesis 1 can be accepted, it needs to be revised.
  4. All the results that test the hypotheses should be improved. The authors should also indicate with which samples and time they are made. It is not understood what they have done or what results they have obtained.

Discusion Review Points

Discussion of results and proposal new studies is missing. The authors have included the subsection and have commented the contributions and limitations. However, they have not 'discussed' the results or proposed future studies.

Author Response

Point1: This manuscript reports on an interesting study that explores charismatic leadership and unethical pro-organizational behavior. I applaud the aspirations represented in this paper. However, both formal and content aspects of the manuscript must be revised. I hope the suggestions I give below will support you in advancing your research efforts on this topic. Following are my specific comments on this paper.

Response: Thank you very much for your very constructive review. We appreciate the time and effort you have committed to reviewing our paper. We truly believe that our revised paper has improved substantially because of your helpful comments.

Point 2: There is a clear statement of the purpose in the introduction. However, the information should be improved.

Response:

Thanks for asking this question and prompting us to explain our argument better. We try to make the arguments clearer in the Introduction section in the updated manuscript. The revised paragraph is as follows (p.2):

“This study answers the call to examine a comprehensive model linking charismatic leadership to unethical behavior. By applying social information processing to the context of UPB, we try to examine the mediated link between charismatic leadership and employee cognition and behaviors, aiming to make at least four salient contributions.”

Point 3: The aim of the study requires a more detailed analysis of the literature on the subject to justify the theoretical model they propose. In addition, it also needs to develop more hypotheses.

Response: Thanks for prompting us to explain our theoretical arguments better. We have added theoretical arguments and developed Hypothesis 3 and in the updated manuscript. The revised paragraph is as follows (p.6):

Hypothesis 3: Performance pressure will moderate the relationship between psychological safety and UPB, such that the relationship will be stronger when performance pressure is higher, rather than lower.

Taken together, it is logical to predict that psychological safety will conditionally influence the strength of the indirect relationship between charismatic leadership and UPB. It is expected that high-performance pressure strengthens the indirect relationship by enhancing the mediating effect of psychological safety between charismatic leadership and UPB. Hence, by facilitating psychotically safety, employees with high-performance pressure tend to engage in UPB under the supervision of charismatic leaders. Thus, we propose a moderated mediation model to explain the effect of charismatic leadership on employees’ UPB. Thus, we hypothesize that:

Hypothesis 4: Performance pressure will moderate the indirect relationship between charismatic leadership and UPB through psychological safety, such that the relationship will be stronger when performance pressure is higher, rather than lower.”

Point 4: The sample works in only one company. Is it a case study? I think that the study has a questionable sampling method and need to be improved.

Response: Thanks for providing us an opportunity to clarify your concerns. We have revised the Participants and Procedure section. The revised paragraph is as follows (p.6-7):

“This study was performed in a large service company located in Ji Lin city, northeast China. This company with about 1000 employees that providing skiing services to tourists. The CEO and human resource department distributed research announcements to employees. The participants in this study comprised 300 full-time working adults who were selected at random from a list of employees in the company. Service industry presents some uniqueness interactions between employees and customers that makes it particularly suitable for our research purpose. Previous research has demonstrated that such workplace environment is for examining third-party oriented behaviors and UPB [9,68]. Thus, we selected this sample to test our hypotheses by focusing on a single industry in China context. In order to avoid common method bias, this study adopted a multi-period data collection method where participants were asked to answer the questionnaires throughout the whole February in 2019. Previous research suggests that time separation between predictor and criterion variables can reduce common method variance bias by decreasing consistency motifs and demand characteristics [69]. Thus, we used three separate surveys with intervals of two weeks between each survey. A time lag of two weeks is consistent with published work examining the influence of leadership on UPB [70].”

Point 5: Appropriate identification of participants is critical and this study. This paper does not adequately described participants. Moreover, the authors should describe each of the samples or, if they do intrasubject analyses, they should include only the participants who are at all times.

Response: Thanks for asking this question and prompting us to explain our argument better. We have revised the Participants and Procedure section. The revised paragraph is as follows (p.7):

“Among them, 9 did not provide complete response in the whole study. Furthermore, the respondents who incorrectly responded to the items were removed from the analysis, with 214 participants remaining.”

Point 6: The authors should detail the type of sampling, why this type of sampling is the best, what risks it has for its validity and how they have been solved so that they do not affect the result obtained.

Response: Thanks for asking this question, we have revised the Participants and Procedure section to make it clear. The revised paragraph is as follows (p.7):

“We collected data from eight departments and relied on convenience and snowball sampling. In the first survey (T1), with the help of CEO and human resource manager (HR), we asked followers to fill in the questionnaire about gender, age, education, work tenure and charismatic leadership. Two weeks later, at Time 2, participants rated psychological safety. Two weeks after this, at Time 3, respondents provided ratings on UPB. There were 276 respondents in the Time 1 survey (92% response rate), 232 participants in the Time 2 survey (84.05% reaction rate), and 226 respondents in the Time 3 survey (97.41% reaction rate). Among them, 9 did not provide complete response in the whole study. Furthermore, the respondents who incorrectly responded to the items were removed from the analysis, with 214 participants remaining. Of these, 62.62% were men and 46.26% held a bachelor’s degree. Most participants were aged 25 to 30 years (43.46%), with 46.26% having more than three years of work experience.”

Point 7: The scales are not correctly described.

Response: Thanks for encouraging us to strengthen our arguments. We have revised the scales in the Measurements section in the updated manuscript (P.7).

Point 8: The control variables are not justified in the introduction. Moreover, there are no hypotheses about them.

Response: Thanks for your suggestion. We revised the manuscript by explain more about the control variables in the Measurement section rather than in the introduction. Because the effects of control variables are not our main research purpose, we do not propose the effects of controls variables following previous related research. The updated manuscript as follows (p.8):

“Control variables: Based on the recommendations of previous studies [21], demographic characteristics may exert influences on participants’ propensity to engage in UPB. We age, gender, education and work tenure to control for their potentially spurious effects as the findings of Kish-Gephart et al. revealed a week correlation between gender and age and unethical actions [74]. Thus, information on followers’ gender, age, education, and work tenure was controlled.”

Point 9: There is not subsection: ‘data analysis’.

Response: Thanks for your suggestion. We revised the data analysis section in the updated manuscript as follows (p.8).

“4.2. Data Analysis

The analyses were modeled in SPSS 20.0 using hierarchical regression. Table 1 presents the means, standard deviations, and correlations among the researched variables. Consistent with the predictions, charismatic leadership is positive related to psychological safety (γ = 0.16, p < 0.01). In addition, psychological safety is positively associated with UPB (γ = 0.20, p < 0.01).”

Point 10: It doesn't explain what the test of measurement models is for. Moreover, it does not indicate with which scales and in which time it has been carried out. It presents some results that we do not know what they correspond to.

Response: Thanks for providing us an opportunity to clarify your concerns. We have revised the scales in the Results section in the updated manuscript (P.8).

“A range of confirmatory factor analyses (CFA) was conducted using Mplus 7.4 to verify how distinctive the variables would be [75]. Four indexed were used to evaluate the goodness of fit: the chi-square statistic (χ2), comparative fit index (CFI), Tucker-Lewis index (TLI), and the root mean square error of approximation with associated 90% confidence intervals (RMSEA). CFI≥0.90 and RMSEA ≤ 0.06 indicate a model’s acceptable fit to the data.”

Point 11: The authors should review all the results. There are many inaccuracies. For example, include the mean and standard deviation of gender and education. Both are categorical variables.

Response: Thanks for pointing out this issue. We added the note to explain the details of these categorical variables to make it clear. The revised paragraph is as follows (p.8):

 “Note:a 1 = male; 2 = female.

     b 1 = less than 25 year old; 2 = 26-35 year old; 3 = 36-45 year old; 4 = 46-55 year old; 5 = 56 year old.

c 1 = below high school; 2 = Junior college; 3 = bachelor; 4 = master;

d 1 = less than 1 year; 2 = 1-2 year; 3 = 2-3 year; 4 = 3-4 year; 5 = more than 5 year.”

Point 12: In addition, the correlations are very low and practically non-existent. This means that perhaps the hypotheses are not worth testing. For example, the correlation between charismatic leadership and psychological safety is 0.16. This is a very low positive correlation. Therefore, even if hypothesis 1 can be accepted, it needs to be revised.

Response: Thanks for providing us an opportunity to clarify your concerns. Although the correlation between charismatic leadership and psychological safety is weak, it does not mean there is not a relation. P value is less than 0.05, which shows that the correlation is existence and meaningful.

Point 13: All the results that test the hypotheses should be improved. The authors should also indicate with which samples and time they are made. It is not understood what they have done or what results they have obtained.

Response: Thanks for providing us an opportunity to clarify your concerns. we have revised the Participants and Procedure section to make it clear. The revised paragraph is as follows (p.7):

“We collected data from eight departments and relied on convenience and snowball sampling. In the first survey (T1), with the help of CEO and human resource manager (HR), we asked followers to fill in the questionnaire about gender, age, education, work tenure and charismatic leadership. Two weeks later, at Time 2, participants rated psychological safety. Two weeks after this, at Time 3, respondents provided ratings on UPB. There were 276 respondents in the Time 1 survey (92% response rate), 232 participants in the Time 2 survey (84.05% reaction rate), and 226 respondents in the Time 3 survey (97.41% reaction rate). Among them, 9 did not provide complete response in the whole study. Furthermore, the respondents who incorrectly responded to the items were removed from the analysis, with 214 participants remaining. Of these, 62.62% were men and 46.26% held a bachelor’s degree. Most participants were aged 25 to 30 years (43.46%), with 46.26% having more than three years of work experience.”

Point 14: Discussion Review Points. Discussion of results and proposal new studies is missing. The authors have included the subsection and have commented the contributions and limitations. However, they have not 'discussed' the results or proposed future studies.

Response: Thank you for your positive feedback, and we appreciate your constructive comments and suggestions. We hope that you find the current manuscript is much improved now as a result of the revision.

Round 2

Reviewer 2 Report

Introduction Review Points

  • The authors do not improve the introduction.
  • The authors do not develop more hypothesis.

Method Review Points

  • A number of researchers have investigated the organizational cultural factors that contribute to leadership practices and its behaviors. Then, if the authors analized one organization they should not generalize them.
  • The authors report that they have drawn the sample from 8 departments. They do not describe how much sample they had from each department. Furthermore, they indicate different numbers of people at different times. Do all the participants in each sample use it? They claim that it is a single sample but have used multi-period to avoid common method bias. They should describe the final sample.
  • The scales are not correctly described. For example, I do not know if The Charismatic Leadership scale have one or more factors. Moreover, they do not explain the Farh's scale and how they select the three items and whether the items are valid by themselves.
  • There are not subsection ‘data analysis’ in methodology
